# Application of Structural Entropy and Spatial Filling Factor in Colonoscopy Image Classification

**DOI:** 10.3390/e23080936

**Published:** 2021-07-22

**Authors:** Brigita Sziová, Szilvia Nagy, Zoltán Fazekas

**Affiliations:** 1Department of Computer Science, Széchenyi István University, Egyetem tér 1, H-9026 Gyor, Hungary; 2Department of Telecommunications, Széchenyi István University, Egyetem tér 1, H-9026 Gyor, Hungary; nagysz@sze.hu; 3Institute for Computer Science and Control (SZTAKI), Eötvös Loránd Research Network (ELKH), 13-17 Kende utca, H-1111 Budapest, Hungary; zoltan.fazekas@sztaki.hu

**Keywords:** computer-aided diagnostics, colour spaces, colonoscopy, Rényi entropies, structural entropy, fuzzy classification

## Abstract

For finding colorectal polyps the standard method relies on the techniques and devices of colonoscopy and the medical expertise of the gastroenterologist. In case of images acquired through colonoscopes the automatic segmentation of the polyps from their environment (i.e., from the bowel wall) is an essential task within computer aided diagnosis system development. As the number of the publicly available polyp images in various databases is still rather limited, it is important to develop metaheuristic methods, such as fuzzy inference methods, along with the deep learning algorithms to improve and validate detection and classification techniques. In the present manuscript firstly a fuzzy rule set is generated and validated. The former process is based on a statistical approach and makes use of histograms of the antecedents. Secondly, a method for selecting relevant antecedent variables is presented. The selection is based on the comparision of the histograms computed from the measured values for the training set. Then the inclusion of the Rényi-entropy-based structural entropy and the spatial filling factor into the set of input variables is proposed and assessed. The beneficial effect of including the mentioned structural entropy of the entropies from the hue and saturation (H and S) colour channels resulted in 65% true positive and 60% true negative rate of the classification for an advantageously selected set of antecedents when working with HSV images.

## 1. Introduction

Colorectal polyps and cancer are the third most common disease in the world nowadays [1]. The colorectal polyps–and thus the cancer developing from them–grow on the surface of the colon/rectum. The literature distinguish several types of polyps. The five most common polyps types are the inflammatory, the hyperplastic, the adenomatous, the villous or tubulovillous adenoma, and the serrated adenoma. Structurally and shape-wise two different fundamental forms can be distinguished within these types, these forms are the pedunculated and sessile forms. The pedunculated polyps hang from a bowel wall into the internal space of the bowel. Sessile polyps grow directly on the bowel wall and are often quite flat.

A portion of these colorectal polyps progress into colorectal cancer over time: approximately 10% of all these lesions [1]. This indicates that the colorectal polyp screening is extremely important as colorectal cancer affects a large population, especially the population over age 50.

Medical image processing systems help doctors in preparing their image-based diagnoses. The medical image processing procedures and methods do not make a diagnosis themselves and it is clear that they will not do so in the near future. Their present goal is to assist the doctors by drawing their attention to image–and thus body–areas which are possibly problematic. These methods are referred to as computer-aided diagnosis (CAD) methods. As the colon cancer and the colorectal cancer are among the leading causes of death worldwide, many countries carry out screening for as wide a segment of the population aged 50 and over years as possible. Nowadays, however, in most countries, early-stage screening is not extensive enough, so it does not have a satisfactory impact in the prevention of the disease. The colorectal polyps can be detected with a special endoscope called colonoscope and also with a computer tomography device. CT scans do not always provide adequate results, futhermore the doctors cannot take samples or remove polyps during the examination procedure [2,3,4]. Both the CT and the colonoscopic procedures are inconvenient for the patients and require thorough preparations from the patients. In the course of colonoscopy examination the doctor inserts and leads a movable, flexible endoscope into the bowel through the rectum. This device not only has a lighting and recording device but also has small tweezers and loops with which samples may be taken or even lesions can be removed.

Many of the colonoscopes comprise additional light sources for narrow-band imaging (NBI), and a small container with which they can squish some bluish-coloured liquid usually (indigo carmine food colour) to a designated area: this bluish-coloured liquid concentrates in the groves, pits and valleys of the uneven bowel wall, and makes the surface structure more perceptible. This branch of the endoscopy is called chromoendoscopy [5].

Capsule endoscopy is another relatively new alternative to classical colonoscopy, its main drawbacks are that it cannot intervene, and it does not facilitate distance and size estimations [6,7].

In the polyp classification the first step is to determine the shape of the polyp [8,9]. As it has been mentioned above, polyps can be depressed into the bowel wall, i.e., flat, slightly protruding sessile polyps or pedunculated polyps with expressed stalk.

As a second step the investigation of the polyp’s surface is carried out. Based on its texture, the types that are likely to be of cancerous nature can usually be distinguished from the non-cancerous polyp surfaces. The pit pattern analysis was developed by Kudo and his coworkers for this purpose. They proposed a five-grade system based on the polyps surface texture on magnified colonoscopy images [10,11,12]. Those polyps surface textures that have almost regular, perfect circular pits of neither very dense and not too sparse spatial density and have regularly distributed pattern, they are the benign polyps. These polyps constitute the class A1. If the surface of the polyp has a pattern like a star, it is light pink, and has regular arrangement, then these polyps–similarly to the A1 polyps–do not tend to develop into malignancy. These polyps belong to the class A2. However, if the surface of the polyp has tubular shapes (class B3L) or extremely small round pits with high spatial density (class B3S), then those polyps may potentially develop into carcinoma. These usually show some characteristic discoloration, at especially in NBI [5]. The class B4 consist of polyps that are dark and have elongated pits, or have meander-like pits-trenches. These have already become cancerous. Those polyps that have irregularly textured surface of black or whitish colour, are in the advanced stadium of cancer, these constitute class B5.

We note here that the question, whether a polyp is malignant or it can turn into a malignant tumor, can only be answered unambiguously by biopsy, followed by a histological examination.

Bernal and his coworkers developed a polyp detection method based on the general shapes and colour of the polyps [13]. They have set up and organized an international scientific challenge, i.e., a scientific competition, Medical Image Computing and Computer-Assisted Intervention (MICCAI) for image segmentation and polyp detection [14]. They prepared three still image databases with different emphasis for this scholarly competition. These databases are called the CVC-Colon, Etis-Larib, and CVC-Clinic databases. In these a specific binary mask is associated with each image. The mask specifies the shape(s) and location(s) of the polyp(s) in the image. The mentioned masks are drawn by human medical experts, thus–due to the limited annotation time–they might be inaccurate. Further causes of the inaccuracy are the manual processing, and the possible vagueness of the polyp perimeter (i.e., due to occlusion or suboptimal lighting conditions). In the aforementioned challenge a wide range of algorithmic methods had been applied by the competing teams, from convolutional neural network coupled with fully automated learning to various hybrid polyp detection methods. There was also absolutely handcrafted method that was based on the idealized appearances of the polyps. As in medical image processing the main issue is the low availability and high cost associated, therefore we decided to use a dedicated method that relies on fuzzy inference.

Fuzzy sets are a conceptual extension of the Boolean sets (where any object can have either 0 or 1 membership values in the sets): In 1965 Zadeh introduced memberships that have real values between 0 and 1 [15,16]. Using fuzzy memberships instead of the classical “yes” and “no” values, the “if $A_{1}$ and $A_{2}$ and …, then $B_{i}$”-type rules can be turned into more flexible and more applicable rules and associated algorithms, e.g., for decision and control. The fuzzy rules, however have to be determined, as also the membership functions of each antecedent $A_{j}$ for each of the consequents $B_{i}$, thus the use of a training set is also necessary for fuzzy decision. If the number of training pictures is not very large, then the rules can be adjusted, made more efficient by incorporating expert knowledge. These two aspects justify the application of fuzzy inference for the purpose.

Also a technique called fuzzy rule interpolation [17] can be applied to images, for those cases when some of the antecedent parameters have no membership at any of the rules, and therefore a classical Mamdani-Assilian [18] or other commonly used inference method will not give valid output.

Based on our research interest and experience in fuzzy classification, we took an fuzzy inference based approach for detecting and classifying colonorectal polyps and developed a method that classifies image segments according to the inferred presence of one or more polyp being shown in them [19]. The aim is not a perfect segmentation of a polyp within a colonoscopic image, but to draw attention of the medical personnel to a given image segment or to several image segments, where polyps might be visible. Firstly, the images are cut into smaller segments, and ech segments is labeled with a fuzzy decision value. Initially, we computed and used certain statistical parameters of the image segments and fitted polynomials to these [20]. However, polynomials introduced rather large sensitivity–in respect of the tile-size–into the method, so we dismissed this approach, but instead applied an edge-sensitive approach [21]. Later different preprocessing methods were incorporated into the approach and their influence were also studied [22].

After some experimentation it has become evident, that a general rulebase that is derived from the whole image-base may not be effective, as there are many different types of polyps furthermore, these polyps had been photographed from fairly different angles and viewpoints, not to mention; the problem associated with the different state of cleansing, in which the bowels appear in the pictures. For these reasons grouping process we made sure to include within each group only images ta, we grouped the images within the databases into image groups. During this grouping we made sure to include within each group only images taken from fairly similar angles of the polyps. These different image groups resulted in different classification rulebases, however the performance of other rulebases were also tested on each image group. Nevertheless, many of the heterologous rulebases showed similar efficiency as the homologous ones, therefore, these could be joined or utilized in the construction of a joint rulebase, or of a multilevel fuzzy classification scheme.

Analysing the histograms of the factual (measured and computed) data for each of the antecedents, it was realized that the fuzzy membership functions corresponding to “polyp” and “no polyp” outputs were very similar for certain antecedents, suggesting that by omiting them all, or at least some of these antecedents would be a viable option without much contraproductive effect on the result.

Bernal and his coworkers [23,24,25] carried out a studies, in which they tracked, where the human experts’ eyes were looking in the images during polyp searches, they found that the circular or similarly shaped shadows, i.e., intensity valleys and intensity gradients, that are characteristic to hemispheres, had been the foci that drew the experts’ attention, and therefore, such shadows are considered the key factors in the localization, identification, recognition, characterization and classification of polyps. Guided by this study we selected such statistical features of images that are closely linked to 3D shapes like hemispheres and spherical calottes [26]. In conjuction with the image quality perceived by the experts, it turned out to be beneficial to compensate for the poorly illuminated environment by transformation from RGB to HSV colour space and increasing the V values [27]. In the work presented herein, we also rely on the HSV colour channels furthermore, this colour space have become an aspect of the investigation.

In the following conciderations, the effects of the above factors are analyzed, namely, the advantages and disadvantages of the mentioned colour spaces, the selection of the input parameters, and reduction of their number. In Section 2, the antecedents are listed and motivated, with special attention to the Rényi entropy antecedents. Next, in Section 3 the mathematical background of the fuzzy inference and fuzzy rule interpolation is summarized for the Readers’ convenience. In Section 4 the results are presented; and finally, in Section 5, the conclusions are drawn and our plans for further research in the area are outlined.

## 2. Antecedents

Colonoscopy images used in the present research are from the three databases published during the MICCAI endoscopic vision challenge. This international scientific challenge was dicsussed above in the Introduction. These databases are the CVC-Colon, Etis-Larib, and CVC-Clinic databases. These consist of 380, 195, and 612 colorectal endoscopic images, respectively. The images are of different size and resolution, the image sizes are 574 × 500, 1225 × 966, and 384 × 288 pixels, respectively, while the resolutions are 72, 72, and 96 dpi, respectively. All three databases have a resolution of 24-bit in the RGB colour space with the primary colours sharing equally the bits. Only the first database uses some low-level compression, the other databases store and manage uncompressed images. For each image in the databases, a binary mask had been manually created to specify the position of polyps in the image. This makes the databases suitable for training learning algorithms, as well as other intelligent methods; in the particular case a fuzzy classification scheme.

Analysing characteristics of the areas where polyps were present, in the first step, we partitioned the images into square tiles. Our primary purpose was to develop a classification algorithm that can establish if the image segment contains polyp or not. This decision is reached by means of a fuzzy classification system. The antecedents were selected as described below, in Section 2.1.

### 2.1. Average, Standard Deviation, Gradients

In colonoscopy, the endoscopic camera takes snapshots in an originally dark, wet and mucous environment lit by a number of point-like light sources. For this reason, the pictures usually contain lots of reflections. Moreover, as the bowel wall is originally pink, the colour range of these pictures is rather narrow (except for the cases, when either the bowel is not cleansed sufficiently, or the lesion present in the image has turned malignant). The input parameters for the fuzzy classification system need to be selected for such an unfriendly spatial, biological, optical environment considering the mentioned colour composition.

In [23] the images were classified into three classes based on the angle of the view of the polyp: zenithal, lateral, and semilateral, see Figure 1. In the zenital case, the polyps can be seen in a top view, from this view they are not significantly lighter than the environment, but their circumference and the surface texture are usually well distinguishable. A polyp when viewed from its side (i.e., from a lateral view) with a much darker–distant and therefore, worse lit–bowel wall background. Then there is also an intermediate semilateral view. In this case, the background is somewhat darker than the polyp but often the contour, especially at the polyp’s base’ is hardly visible.

The polyps are usually protruding from the bowel wall, thus they are lighter in the images than their surroundings. Based on the above viewing-direction-dependant visual attributes of the polyps, the average pixel intensity, the standard deviation, the mean of the gradients’ absolute values, the standard deviation of gradients magnitudes were chosen as antecedents, as the gradients around the polyp-blobs can be quite large.

### 2.2. Edge Density

If the edges are strong and distinct (i.e., in most of the zenithal and lateral pictures and in some of the semilateral ones) edge images provide much information on the location of the polyp. When computing the edge density within a picture segment, firstly an edge detection–using an edge detection filter–is carried out.

For instance, a Canny filter making use of the Sobel matrices $S_{x}$ and $S_{y}$ can be emploiyed to select pixels where a rapid change in intensity occurs [28].
(1)$$S_{x}=\left(\begin{array}{ccc} -1 & 0 & 1\\ -2 & 0 & 2\\ -1 & 0 & 1 \end{array}\right.\;\mathrm{and}\;S_{y}=\left(\begin{array}{ccc} -1 & -2 & -1\\ 0 & 0 & 0\\ 1 & 2 & 1 \end{array}\right)$$

As a simple measurable quantity, the ratio of the edges within the image segment was calculated. If there are many edges in the picture segment (e.g., due to veins, polyp texture, bowel wall folding, polyp contours), this value will be high, for smoother image segments this value tends to be low.

### 2.3. The Rényi Entropy Based Structural Entropy and the Spatial Filling Factor

In image processing and computer vision, entropies are proven to be useful features [29]. Beside the usual von Neumann [30] and Shannon [31] entropies, there are many other generalizations and these are also in use in the mentioned disciplines [32,33]. There are a number of algorithmic approaches that apply some sort of enropies to images, but they all need a conversion from the pixel intensities to probability distributions, i.e., firstly, intensity histograms need to be computed for the images and then these histograms need to be normalized. The entropy is defined by the probabilities of the elements of a set,
(2)$$H\left(S\right)=-\sum_{i=1}^{N}p_{i}\log p_{i},$$
with *H* being the entropy of set *S* where the probability of the *i*th element’s occurance is $p_{i}$ and the total number of these elements is *N*, e.g., in the above example *N* = 256. In other approaches, certain neighbourhoods are looked at and their probability distributions are derived, or even differences between corresponding elements of the neighbourhoods are analysed in this manner. As pixel intensities relate to each other within neighbourhood, and the differences of the neighbouring pixels are rather easy to calculate, a possible approach to the “image-to-probability” task is to calculate the probability distribution of these differences [34]. This results in a one-dimensional (i.e., neighbour difference) distribution. It can be extended to 2D, if *x* and *y* directions are considered separately. Moreover, many other dimensions could be introduced if greater (i.e., not just direct) neighborhoods are to be considered.

Another, straightforward “image-to-probability distribution” conversion possibility is to normalize the complete image (or a part of it) so that the sum of the pixel intensities would be 1, thus they could be viewed as a 2D probability distribution [35]. Pipek and Varga [36] introduced two entropy related quantities, that can give a quantitative measure of the overall shape. One of the quantities, the logarithm of the spatial filling factor *q*,
(3)$$\log q=S_{2}-S_{0}$$
was already used in quantum physics. Here $S_{2}$ and $S_{0}$ are the 2nd and 0th generalized Rényi entropies, that can be calculated according to the following formula,
(4)$$S_{n}=\frac{1}{1-n}\log\sum_{i=1}^{N}p_{i}^{n},$$
where *N* is the number of pixels. The probability $p_{i}$ is the normalized pixel intensity,
(5)$$p_{i}=I_{i}/\sum_{j=1}^{N}I_{j},$$
with $I_{j}$ being *j*th pixel’s intensity value. For the limit n→1 the Formula (Equation 4) results in the classical Shannon entropy.

The other quantity is the difference of the total entropy $S_{1}$ and the entropy that is associated with the ratio of the higher probability points, which is the 2nd Rényi entropy. This quantity was named structural entropy, and can be calculated as
(6)$$S_{str}=S_{1}-S_{2}.$$

The position of the probability distribution on the $S_{str}\left(\log q\right)$ map gives information about the localization, shape of the distribution. Various distribution shapes, like Gaussian, spherical, sinusoidal or power law distributions have different trend lines on the $S_{str}\left(\log q\right)$ map, see Figure 2, for instance, if a probability distribution is an exponential function, it will have an exponential trend line. For further details and the connection between the physical quantities and the entropies, and the trend lines associated with various distribution shapes, see [37]. In image processing, based on the shape of the trend lines associated with the
(7)$$p\left(x_{i},y_{j}\right)=A\cdot e^{-|B\cdot x_{i}+C\cdot y_{j}{|}^{\alpha }}$$

Bonyár introduced the notion of the localisation factor α [38], which he used for characterising roughness of micro and nano surfaces [39]. The above Rényi entropy based quantities are used in detecting the shape and the pattern on the surface of the polyp [29]. We found significant difference between the responses of corrugated bowel wall and the semisphere-like polyps with or without shadow. For this reason, we introduced the $S_{str}$ and logq quantities as antecedents.

### 2.4. Wavelets

Wavelet transformation is effective tool widely used in signal and image processing, if different scale behaviour need to be identified or studied. A wavelet transform generates the smoothed versions of the studied signal as well as the fine details that are necessary to move from the smoothed signal to the original signal. The fine details are usually large at the positions where rapid changes are present in the image. If the signal is 2D then the wavelet transform should be performed in both dimensions. Our colonoscopy image segments are also 2D discrete signals within each of the three colour channels. As around the polyps there are rather rapid variations, we expect the fine scale components to be relatively large near and around the polyps. The above properties served as motivation for us to include wavelet transforms of the image segments in our study. The same parameters were calculated for the wavelet transforms as for the original images themselves. See antecedent indices 16–75 and 82–99 in Table 1.

A 1D discrete wavelet analysis is a set of transformations with the following properties. The original series $x_{n},x=0.1,\ldots$ are expanded into series $c_{n}$ and $d_{n}$ with approximately half of the original length $c_{n}$ and $d_{n}$. The former is computed by expanding the series according to a scaling function $\phi_{i}$, that is
(8)$$c_{\left\lfloor n/2\right\rfloor }=\sum_{i=1}^{N_{s}-1}\phi_{i}x_{n-i}\;\mathrm{if}\;n=2k,$$
where $N_{s}$ is the size of the sliding window and ⌊·⌋ denotes floor operator, i.e., the integer part of the value. Here it is visible, that the length of the series is (approximately) halved. Also, the odd indices *n* are omitted for simplicity from the present paper. This transformation gives sliding-windowed weighted average over a sliding window of lenth $N_{s}$. It characterizes the larger scale behaviour of the discrete signal. Practically, this operation is implemented as a low-pass convolutional filter and then downsampling. The latter series, i.e., $d_{n}$ is the output of a high-pass convolutional filter, that is represents the rapidly varying components of the signal. Similarly to (Equation 8), it can be represented as expansion of series according to a wavelet ψ as
(9)$$d_{\left\lfloor n/2\right\rfloor }=\sum_{i=1}^{N_{s}-1}\psi_{i}x_{n-i}\;\mathrm{if}\;n=2k.$$

In the case of 2D signals, such as images, the wavelet transform is usually carried out separately in both dimensions, thus producing four output pictures, see Figure 3. The low-pass–low-pass output (LL) gives the average behaviour of the image. The second and third, correspond to the mixed filters, that is low-pass–high-pass (LH) and high-pass–low-pass (HL). The respective output images have near-zero pixel intensity in the slowly varying domains, and high intensity, where vertical, or horizontal changes occur in the original image parts are present. The fourth image is the output after the two high-pass filters (HH), it has non-zero pixels where the variations are large between the neighbouring pixels in both directions. These transformed images were analyzed similarly to the original ones.

### 2.5. Initial Set of Antecedents

We generated 99 input parameters including also the above listed measurable quantities. The antecedents (i.e., these 99 input parameters) of the fuzzy rule base are listed in Table 1.

### 2.6. Colour Spaces HSV and RGB

First we carried out research on RGB (red, green, blue) images from the mentioned image-bases, then we tested our approach on images converted to HSV colour space. The RGB colour space is used primarily in computer graphics, and seemed a natural choice. However, the results derived from RGB images were not encouraging. As according to the findings in [27] the H and the S channels of the HSV colour space are suitable for both suppressing and highlighting shadows and this advantageous feature is relied on by many image processing methods in various application areas, we decided to continue our experiments in this colour space.

## 3. Fuzzy Inference and Interpolation

### 3.1. Fuzzy Sets and Inference

As mentioned in the Introduction, the fuzzy sets which had been introduced in the seminal work by Zadeh [15], are set-like constructs, whose members (in case of ordinary sets: elements) can have not only full, membership represented by (binary) 1, but also partial membership values represented by real values between 0 and 1. Such a “fuzzy” approach is used by people in conjunction with everyday notions such as ‘tall’, ‘lean’, and ‘fat’. This is expressed with words like ‘very’, ‘somewhat’ and so on. For example a human can be considered tall with 1 membership value if they are taller than 2 m, but in the case of 1.7 m, they can be considered only with, say, 0.1 or 0.3 membership values. The membership function μ(x) associates membership values with all the possible values *x* of the set *X* being studied, like in our above example the possible human heights. The interval $I_{\alpha }$ of the base set x∈X that arises if the membership function is thresholded at a given membership value α, i.e., the {x:μ(x)≥α} is called the α-cut.

Using fuzzy membership functions as rules for decision makes it possible to classify some measured data with respect to overlapping intervals and regions. Furthermore it makes the decision process more flexible. A statement like “if a man is tall and has slow reaction time then he can not be a successful Formula 1 pilot” could be a rule-of-thumb for the selection of the Formula 1 pilots. However, the separation between being tall and not being tall is far from being distinct. Similar sentences that are understandable for humans and common are not interpretable by computers, using only Boolean logic, such sentences obtain meaning vialy used in everyday talk fuzzy logic. Nonetheless, in order to construe a meaning to the above example, also the logical operators “and” and “or” need to be adapted to the fuzzy logic, i.e., need to be fuzzified. In the fuzzy logic literature, there are a number of different adaptations, definitions of these operators/operations. One of the most widespread is the Zadeh t-norm and s-norms, where the “and” operation between two membership values is their minimum, and the “or” operation is their maximum. See the relevant definitions in [15] and some applications of these in [40,41].

Using these operations complex decisions can be carried out in a straightforward manner. For the purpose of fuzzy inference, for example, Mamdani and Assilian suggested in [18] a fairly simple scheme: the antecedents are fuzzy-logically connected with an “and” while the consequents are connected in the mentioned sense by an “or”; see Figure 4.

In the computer-aided diagnosis application considered herein, the rules that are used in decision are represented by membership functions similar to the ones shown in Figure 4. In this work, a method relying on triangular fuzzy sets and their characteristic points (infima and suprema of the supports and the highest value points) was used. These values were computed from the minimum, maximum and mean of the empirical measured data from the training set [20], later also these triangles were fitted properly to the shape of the training set’s measured data’s histograms [42].

Herein we also apply such fitted membership functions. However, the histograms in some cases have very low numbers outside of a narrow interval, furthermore, in some cases very sparse fuzzy rules arise, with small or no overlap at all between the supports. Such an example is given in the 2nd column of Figure 4. In such cases, i.e., if the measured data falls into a domain where both of the rules have zero membership. Hence, the fuzzy inference can not be carried out. For such cases the fuzzy rule interpolation can help to come up with some meaningful membership values for the purpose of inference.

### 3.2. Fuzzy Rule Interpolation

As it was discussed above there are cases when conventional inference methods do not lead to valid results in the inference, thus fuzzy rule interpolation is necessary. One of the possible interpolation methods was developed by Kóczy and Hirota [17,43]. This method uses the natural concept, that if an observed value (even fuzzy value) is more similar to one of the antecedent rules, then its output will also be more similar the corresponding consequent. To implement this approach, it is necessary to measure the distance of the observed value from the fuzzy sets. In case of a simple triangular membership functions, such a distance can be calculated simply from the characteristic points of the membership functions, i.e., from the vertices of the corresponding triangle. In order to turn this vague idea into a viable practically usable (e.g., stabil) method ita modification was necessary [44]. According to this modified method the characteristic α-cuts (α=0 and α=1) of the conclusion $B^{*}$ based on the antecedent sets $A_{i}$, and consequent sets $B_{j}$ can be calculated as
(10)$$\inf\left\{B_{\alpha }^{*}\right\}=\frac{\sum_{i=1}^{2n}{\left(\frac{1}{d_{\alpha L}\left(A^{*},A_{i}\right)}\right)}^{k}\inf\left\{B_{i\alpha }^{*}\right\}}{\sum_{i=1}^{2n}{\left(\frac{1}{d_{\alpha L}\left(A^{*},A_{i}\right)}\right)}^{k}},$$
and
(11)$$\sup\left\{B_{\alpha }^{*}\right\}=\frac{\sum_{i=1}^{2n}{\left(\frac{1}{d_{\alpha U}\left(A^{*},A_{i}\right)}\right)}^{k}\sup\left\{B_{i\alpha }^{*}\right\}}{\sum_{i=1}^{2n}{\left(\frac{1}{d_{\alpha U}\left(A^{*},A_{i}\right)}\right)}^{k}},$$
if the measured data is $A^{*}$, the number of the antecedents is *k*, and the Euclidean distance operator is denoted by d(·,·). This type of rule interpolation was applied in the present study. Furthermore, also the original, triangular rules were used in a Mamdani-like inference system, as well as three types of membership functions with extended supports. This approach was necessary as the mentioned Kóczy-Hirota interpolation only fills the gaps around the narrow rules that belong to different consequents in a given antecedent dimension.

## 4. Experimental Setup

It has become an accepted development practice usual to test the classification methods to be used on colonoscopy images on public annoted databases and develop and invest in the method further only after these tests are succesful. In [14] many methods and their respective results are listed therefore it is convinient to compare the results arrived to via the methods proposed herein and use the databases that had been provided for the MICCAI colonoscopy challenge mentioned in the Introduction [13,14,24]. Some information has been already given in Section 2 on the image sizes used by the mentioned databases. In line with this image-base oriented approach all the three MICCAI image-bases have been used.

As a first step, each colonoscopy image was cut into roughly the same number of tiles. The tile sizes were chosen in the following manner for ETIS-Larib [24] it was 200 by 200 pixel, for the CVC Colon [23] 100 by 100, while for the CVC Clinic [13,14] 50 by 50. From every second tile a training set was built, and the other half of the tiles was used for testing the derived rulebases. The number of the non-polyp tiles vastly outnumbered those with one or more polyp. There were 13733 from the former, while only 4066 from the latter in the test set respectively. In the two reference cases, we used all the 99 antecedents from Table 1, i.e., for RGB and HSV colour spaces. Later, however, during the experimentation with various antecedent sets some of the antecedents were omitted.

As a second step, the fuzzy rulebases were generated from the data gained from the training set according to the method outlined at the end of Section 3.1. Thus three sets of rules were created, the first corresponds to the triangle’s core at the mean (Figure 5), the second at the median of the measured data, and the third rulebase was fitted to the maximum of the histogram. The suprema and infima of the triangles in this case were set to the first points where the historgram (that is normalized to 1 as its maximal column value) surpasses the 0.01 limit starting form both sides.

As a last step we evaluated the tiles in the test set, and compared their inferred polyp-content to that of the associated image masks which had been drawn by experts. Figure 6 shows the plan of the method that can help in finding polyps and our present work is still a part of the preparatory phase: the selection of the antecedents and building the fuzzy rules.

### 4.1. Using All the Initial Antecedents

If all the initially selected input variables were used in the inference, both for the mean centered triangles and for the median centered ones, the results turned out to be not very encouraging. In the case of mean and median centered types of rulebases the results were practically always “no polyp”. In case of the histogram fitted rulebase, both the true positive and the true negative rates (TPR and TNR) were just marginally better than the the corresponding rates associated with a random classification (TPR=0.9567, TNR=0.1987, MCC=0.18, PPV=0.26, NPV=0.94, κ=0.08). The results in case of HSV were either almost all positive, almost all negative. Furthermore, the results depend on whether interpolation or classical inference was used with extended supports.

### 4.2. Using Only the Rényi Entropy-Based Antecedents in the Rulebase

As the structural entropy and the spatial filling factor seemed to be promising in this type of classification [29], therefore we decided to check, whether using only these variables from the image tiles and their four wavelet transformed versions can lead to more probable decisions. When this approach was followed then the histogram fitted results improved significantly. For the HSV images, the TP and TN rates achieved 55.80% (95% CI [0.5572, 0.5587]) and 65.79% (95% CI [0.6575, 0.6583]), respectively, which means, that these entropies enable a better classification than all the previously selected variables together, even though they are still low, the improvement is visible. The other performance metrics for this case were MCC = 0.1857 (95% CI [0.1850, 0.1864]), PPV = 0.3257 (95% CI [0.3253, 0.3261]), NPV = 0.8341 (95% CI [0.8338, 0.8343]), κ = 0.1726 (95% CI [0.1719, 0.1732]). The 95% confidence intervals (95% CI) were calculated according to formula
(12)$$\left[M-z^{*}\cdot \frac{\sigma }{\sqrt{n}},M+z^{*}\cdot \frac{\sigma }{\sqrt{n}}\right]$$
where *M* is the mean, σ the standard deviation of the measured values, *n* is the number of samples and $z^{*}$ is 1.96 for the confidence level of 95%. The generation of the population was carried out by randomly selecting 80% of the possible test image segments, and the population element number was n=100.

As it can be seen from Figure A1, Figure A2, Figure A3, Figure A4, Figure A5, Figure A6, Figure A7 and Figure A8 in Appendix A, the histograms of structural entropies and the spatial filling factors calculated from the HSV channels i.e., the 1st and 3rd columns of antecedent figures corresponding to the mentioned quantities. According to Table 1, the antecedents concerned were those marked with 10 to 15, 25 to 30, 40 to 45, 55 to 60, and 70 to 75). For the tiles–both–with and without polyps are centered at one side of the value set very near to each other, but the width of the supports for the two membership functions that belong to the two consequents differ significantly.

It seemed to be a reasonable approach to select those antecedents, where the histograms differ as much as possible, and the above finding about the structural entropy and spatial filling factor corroborates this approach. After some consideration we extended this set of antecedents somewhat. This extended set is detailed in the next subsection.

### 4.3. Antecedents Resulting in Considerably Differing Histograms for Images with and without Polyps

In order to select those antecedents where the histograms, and thus the resulting rules differ the most, the total absolute distance between the three characteristic points of the respective approximated histograms (i.e., the distances between the vertices of the purple and the cyan triangles in Figure A1, Figure A2, Figure A3, Figure A4, Figure A5, Figure A6, Figure A7, Figure A8, Figure A9 and Figure A10) were calculated. The input variables where this total distance was larger than 0.2 are as follows (in decreasing total distance order), see Table 2.

For these antecedents, the best result for the RGB colour space was achieved by interpolation, but it still was very poor, namely only 35% TPR and 75% TNR. The other performance metrics were MCC=0.1, PPV=0.29, NPV=0.795, κ=0.35. For the HSV colours the results were even worse, almost all tiles in the test set were listed as negative. Therefore, we investigated the reasons for this situation and found that this set contains all the very narrow triangle rules (like antecedents 31, 48, 61, 63, 84, 94), and after some closer investigation, we found that excluding these antecedents considerably improves the results (40% TPR and 75% TNR), not only for the classical triangular Mamdani-type classification, but even for interpolation and rules with extended support. We tested this rather counter intuitive result for many original sets, and excluding these narrow, and further placed triangle rules always increased the results.

The antecedents with total distance larger than 0.35 are presented in Table 3. This set includes only 13 antecedents from the 99, however, in this case almost all the classification results become negative (in medical sense), omitting the mentioned narrow triangle rules it is still 67.16% TPR (95% CI [0.6708, 0.6725]) and 48.52% TNR (95% CI [0.4848, 0.4857]). The other performance metrics were MCC = 0.1324 (95% CI [0.1316, 0.1332]), PPV = 0.2787 (95% CI [0.2784, 0.2790]), NPV = 0.8327 (95% CI [0.8327, 0.8335]), κ = 0.1049 (95% CI [0.1042, 0.1055]). As this approach did not seem very successful, we decided to use the difference of the centers of the histograms as selection rule.

### 4.4. Antecedents with Histograms Resulting in Considerable Distance between the Histogram Centres for Images with and without Polyps

The antecedents, where the centers of the histograms for the two consequents’ training set were fairly distant from each other, were those identified by the following numbers, see Table 4. According to Table 1 these numbers correspond to the input variables listed in Table 5.

For all the input parameters listed in Table 5 the resulting TPR was 64%, and the TNR 60%. Without the gradients (i.e. excluding last row denoted by ${}^{*}$ in Table 5) 65% and 60% was the TPR and TNR, respectively. The Matthews correlation coefficient was with gradient 0.1999 (95% CI [0.1991, 0.2007]), without gradient it improved slightly to MCC = 0.2119 (95% CI [0.2112, 0.2125]). The predictive rates behaved very similarly; for the case with the gradients PPV = 0.3196 (95% CI [0.3192, 0.3200]), NPV = 0.8488 (95% CI [0.8485, 0.8492]), while for the case without the gradients they are only slightly better, PPV = 0.3247 (95% CI [0.3244, 0.3250]), NPV = 0.8538 (95% CI [0.8535, 0.8540]). The Cohen-κ increase also a little, from κ = 0.1751 (95% CI [0.1744, 0.1758]), to κ = 0.1852 (95% CI [0.1847, 0.1858]).

From this point the results did not improve any further, including all the previously omitted entropies affect the results by less than a percent. The resulting indices are not too high, so our research results are not applicable for real diagnosis. It is clear, that a further search for efficient antecedents is needed, as well as a systematic optimization of the antecedents should be included [45], or perhaps some bacterial or other nature inspired algorithms, it is worth to start in this direction, and whether the Rényi entropy based quantities are applicable for this purpose.

In [13] recently finished colonoscopy challenges the performance of the methods were quite varied, the TPR was between 71.4% and 16.7%, the TNR was between 98.6% and 26.6% and the PPV was between 93.5% and 13.6%. In the more up-to-date review [46] the performances improved to (ETIS-Larib dataset) the TPR was between 54.2% and 23.43%, the PPV was between 70.23% and 25.83%, and (CVC-ColonDB dataset) the TPR was between 60.46% and 12.93%, the PPV was between 76.06% and 37.46%.

## 5. Conclusions

A fuzzy inference method–making use of fuzzy interpolation and histogram estimation–for colorectal polyp detection was proposed in this paper. Among the inputs were the Rényi entropy based structural entropies along with other statistical parameters characterizing the image and its wavelet transforms. The method partitions each image into a number of tiles and suggests whether a given tile contains polyp, or not. In earlier works [19,22] a simple triangular rulebase using the minimum, maximum, and mean or median of the training sets’s empirical data was created, which turned out to be not very effective. Later a histogram fitted triangular rulebase provided much better results, so these types of antecedents were employed herein. Our study focused on the selection of the relevant antecedents.

Intuitively, those antecedents seem to be essential for an effective inference system, where the rules derived from the images with and without a polyp respectively, differ as much as possible. However, we found, that even with rule interpolation, the too narrow supported antecedents have disadvantageous influence on the classification results. The antecedents with larger distance between the α cuts in the membership function corresponding to the positive and negative output rules, but still overlapping supports are the most effective.

Using the mean, standard deviation, edge density and Rényi entropy based structural entropy and spatial filling factor, and the gradients were selected as possible antecedents, however, most of these antecedents, like the means and the gradients had no or had negative influence on the classification results. Edge densities were important only in the case of the mixed wavelet transform outputs.

The Rényi entropy based structural entropy from the H and S colour channels of the original and low-pass wavelet transformed image segment, as well as of the (S) of the high-pass wavelet transform was proven beneficial for the classification, but the others could not improve the 65% TPR and 60% TNR of the classification.The influence of changing the colour channel from RGB to HSV was proven to be positive.

In the future we plan to change the method of prepairing tiles, we would like to introduce overlaps between the tiles as too small polyp percentage is not beneficial for the detection efficiency. Also, The results presented in [29] showed that the sliding windows can uncover such properties that can be used for distinguishing between bowel wall and polyp-like structures.

## Figures and Tables

**Figure 1 entropy-23-00936-f001:**
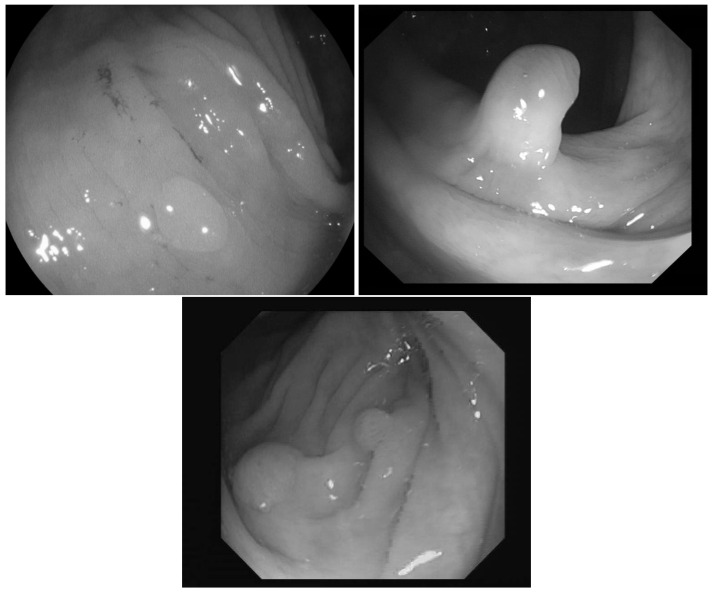
Sample pictures from widely used colorectal image-bases [13,23,24]. The left image shows the zenithal view of a colorectal polyp, the second one is a lateral view, while the third is a semilateral view. The resolution differences of the three databases are also visible on the images.

**Figure 2 entropy-23-00936-f002:**
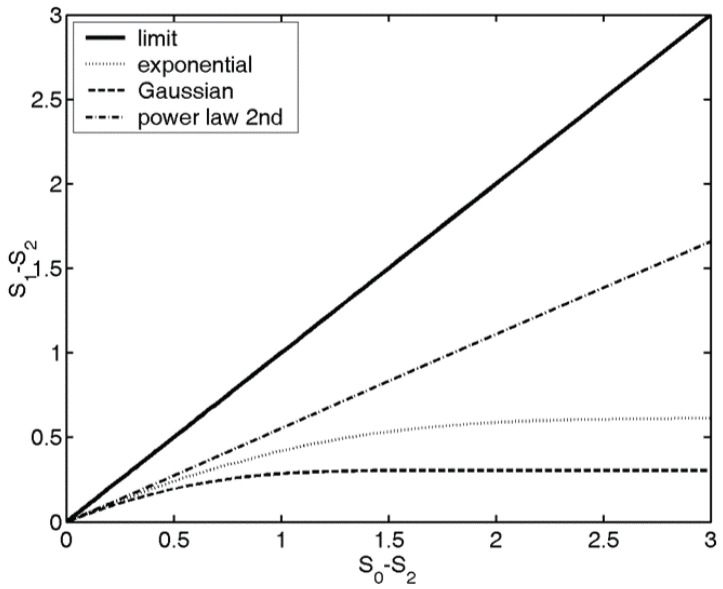
Structural entropy ($S_{Str}=S_{1}-S_{2}$) vs. spatial filling factor (−$\log q=S_{0}-S_{2}$) for various types of probability distributions. The thick solid line shows the theoretical limit: all the distributions have to be below this line, i.e., $S_{Str}\leq \log q$. The other curves show some examples of trend lines.

**Figure 3 entropy-23-00936-f003:**
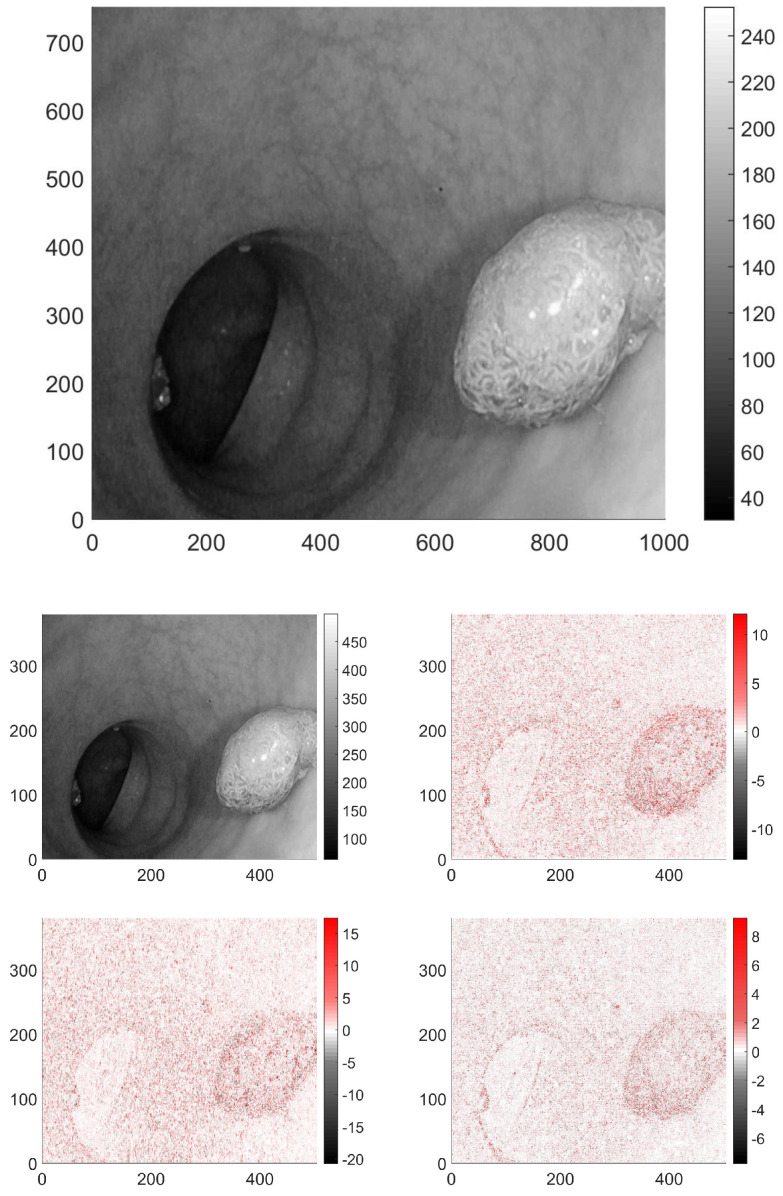
A picture presented in greyscale from database [24], and its respective wavelet transforms. In this case only the LH wavelet transformed version carries perceptable information.

**Figure 4 entropy-23-00936-f004:**
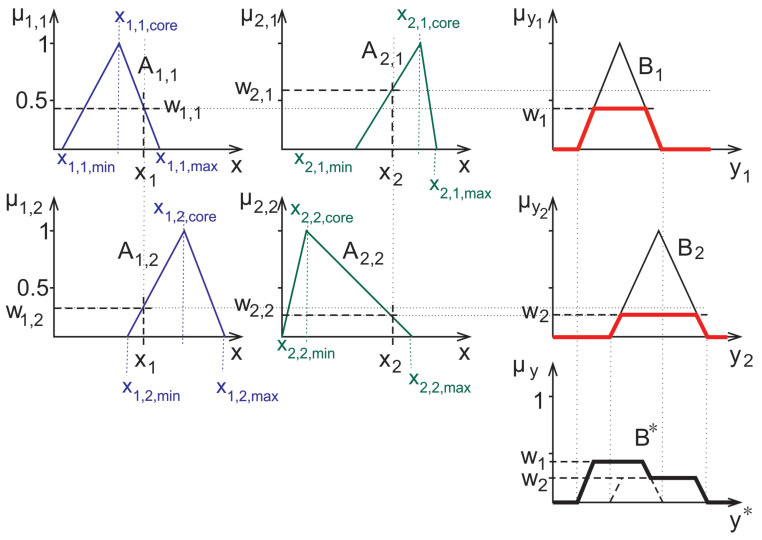
Graphical representation of the Mamadani-Assilian type inference. The antecedents are denoted by $A_{ij}$, their actual measured value by $x_{i}$, their membership value in the rule corresponding to the consequent $B_{j}$ with $w_{ij}$. The defuzzified decision is shown as $B^{*}$.

**Figure 5 entropy-23-00936-f005:**
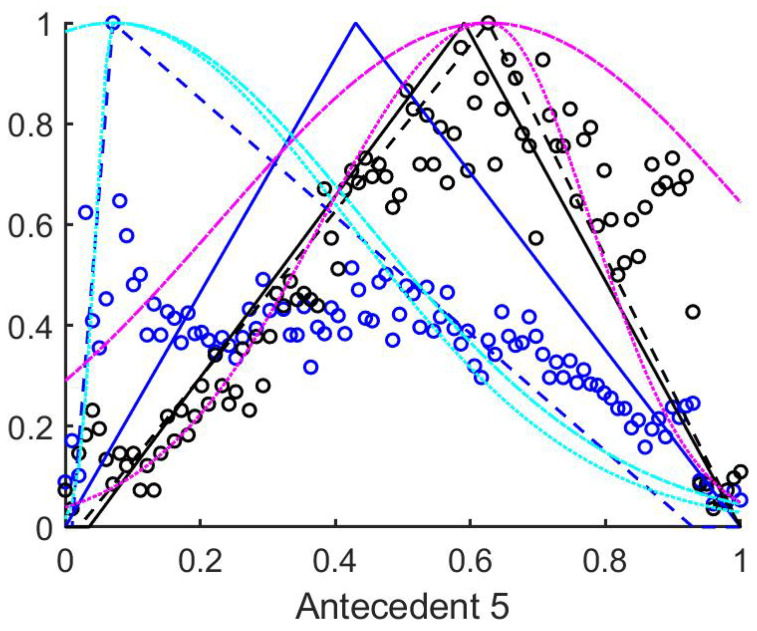
The histograms of a training set of measured data. Black circles mark the no polyp results, while the blue circles the tiles with polyp. The rulebases resulting from all three of the methods are plotted. The black and blue continious poly-lines–used as above–denote the mean centered rules, and the dashed lines the histogram based rules. Also, for the histogram based rules the Gaussian and half Gaussian membership functions with extended support are plotted with cyan and magenta colours. The half Gaussian extended rule membership function consists of two half Gaussians fitted to the asymmetrical triangle membership functions.

**Figure 6 entropy-23-00936-f006:**
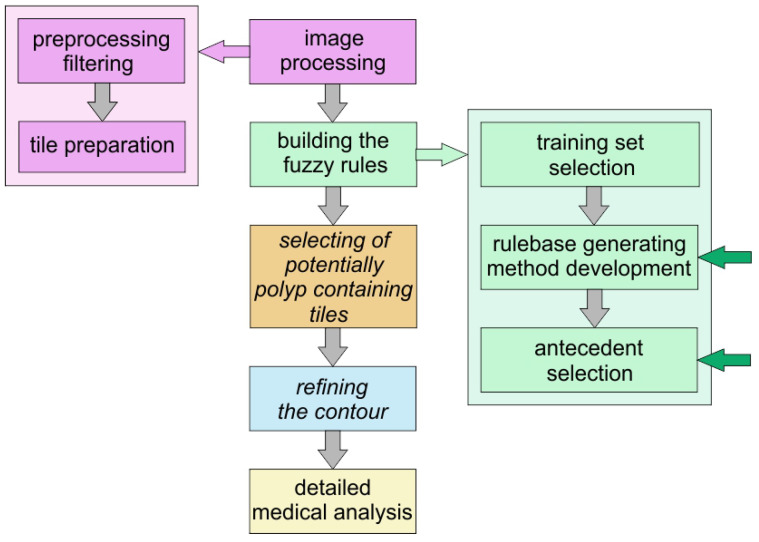
Workflow chart. The first two boxes represent the development of the method, the boxes with italic type letters represent those parts of method which can be automatically performed without human interaction, the others need human control or in the case of the detail medical analysis it should be performed and overseen by human medical experts.

**Table 1 entropy-23-00936-t001:** The indices for the initial antecedent parameters. The indices are given for both the RGB and the HSV colour representations, e.g., antecedents 1 and 2 are the mean and standard deviation of the first colour channel, i.e., of the red (R) and of the hue (H) channel, respectively. The wavelet transformed images are denoted by their respective low-pass and high-pass filters, i.e., LL, LH, HL, and HH.

Index	Antecedent Name
1–2	mean and standard deviation, R/H
3–4	mean and standard deviation, G/S
5–6	mean and standard deviation, B/V
7	edge density, R/H
8	edge density, G/S
9	edge density, B/V
10–11	$S_{str}$, lnq, R/H
12–13	$S_{str}$, lnq, G/S
14–15	$S_{str}$, lnq, B/V
16–30	similar to 1–15, wavelet transform LL
31–45	similar to 1–15, wavelet transform LH
46–60	similar to 1–15, wavelet transform HL
61–75	similar to 1–15, wavelet transform HH
76–77	gradient magnitude’s mean and standard deviation, R/H
78–79	gradient magnitude’s mean and standard deviation, G/S
80–81	gradient magnitude’s mean and standard deviation, B/V
82–87	similar to 76–81, gradient direction
88–93	similar to 76–81, gradient *x* component
94–99	similar to 76–81, gradient *y* component

**Table 2 entropy-23-00936-t002:** Antecedents where this total distance was larger than 0.2 in decreasing order of the total distance.

51, 4, 61, 19, 85, 29, 73, 27, 35, 34, 32, 82, 97, 43, 72, 64, 13, 12, 94, 28, 36, 99, 78, 49, 84, 91,
31, 48, 80, 63, 79, 45, 74, 75, 44

**Table 3 entropy-23-00936-t003:** The antecedents with total distance larger than 0.35.

12, 28, 36, 44, 45, 49, 74, 75, 78, 79, 80, 91, 99

**Table 4 entropy-23-00936-t004:** Antecedents, where the centers of the histograms for the two consequents’ training set were fairly distant from each other.

2, 3, 5, 10, 11, 12, 13, 17, 18, 20, 25, 26, 27, 28, 37, 38, 39, 52,
53, 54, 62, 63, 64, 68, 69, 72, 73, 77, 79, 83, 85, 86, 89, 91, 95, 97

**Table 5 entropy-23-00936-t005:** The antecedents of the reduced set. The colour channels were denoted the following way: H: hue, S: saturation, V: value or intensity. The last group containts the gradients. As to different cases were studied: one, which included these gradients, the other which did not, the last row was denoted by *.

Antecedent Group	Antecedent
Original tile	standard deviation H mean of S and V structural entropy of H and S lnq of H and S
Low-pass–low-pass wavelet transform	edge density of H, S and V
Low-pass–high-pass wavelet transform	edge density of H, S and V
High-pass–low-pass wavelet transform	edge density of H, S and V
High-pass–high-pass wavelet transform	standard deviation of H and S edge densities of S and V structural entropies of S
(Gradients) *	(standard deviation of H and S)

## Data Availability

We used anonymized images from databases that are public and openly available at http://www.cvc.uab.es/CVC-Colon/index.php/databases/ (accessed on 26 April 2021), and https://polyp.grand-challenge.org/EtisLarib/ (accessed on 26 April 2021).

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
