# Peer review of "Application of Structural Entropy and Spatial Filling Factor in Colonoscopy Image Classification"

_entropy, 2021, doi:10.3390/e23080936_

Round 1
Reviewer 1 Report
In this paper, an image classification is proposed. However, there are many issues to improve as a rigorous scientific report,
- Self Citation to the authors previous work which is not related to the research question must be removed.
- There are many figures with low quality. One of which does not even have a caption.
- A lot of different and rather not suitable methods, were discussed in the paper. Such a method is suitable for dissertation not a paper. The most suitable methods must be presented and discussed.
- As a clinical diagnosis method, standards such as stard and tripod must be used, for example the CI 95% of the performance indices must be reported. The abstract must contain important quantitative results.
- There are important indices missing in the results, precision, NPV, MCC, Kappa and AUC.
- Proper feature selection must be used to show which features are informative. Statistical feature selection is preferred.
- Due to the correlation between frames, and the importance of classifying new frame in a new subject, leave one out subject cross validation is required.
- Comparison with the state of the art is missing.
- Sensitivity analysis of free parameters is required.
Author Response
Response to the comments of Reviewer 1
Firstly, we would like to express our sincere thanks to Referee 1 for his/her insightful and helpful comments. We do hope that we managed to improve the quality of our manuscript via addressing his/her concerns.
The following points and concerns were addressed in the new version of our manuscript.
- Self-citation to the authors previous work which is not related to the research question must be removed.
We removed the self-citation from the References. Indeed, the topic covered in cited paper was not directly related to the present subject. In the original version, it was included as it served for us as a motivation, and in our minds it was related to the history of this research.
- There are many figures with low quality. One of which does not even have a caption.
We have included in the new version of our manuscript more readable figures. We have now more detailed and informative captions associated with the figures. Furthermore, some new figures help to orient the readers in the field (e.g., Figures 1., Figure 2., Figure 3. ), and some (Figures 4., Figure 5.) clarify certain notations used.
- A lot of different and rather not suitable methods, were discussed in the paper. Such a method is suitable for dissertation not a paper. The most suitable methods must be presented and discussed.
In this new version, we focus on the histogram-based rule generation for the method. We cite some more relevant and more recent papers (e.g., [46]) and their methods to facilitate the comparison.
- As a clinical diagnosis method is being presented in the manuscript, standards, such as STARD and TRIPOD, must be used, e.g., the CI 95% of the performance indices must be reported. The abstract must contain important quantitative results.
We inserted some quantitative results also into the abstract. We studied TRIPOD and followed several of its recommendations (e.g., rewriting the abstract, so that it would reflect more the second point of the TRIPOD, specifying the objectives more clearly, comparing multiple models, giving more information about the use the database.). We hope that in this way we managed to sufficiently improve the presentation quality of our research. By now, we do understand and appreciate the importance of this reporting practice in clinical diagnosis. Indeed, in our view similar practice should be followed in other fields of science, as well.
Our method presented in the manuscript is ’under construction’, it is still being improved (e.g., by using overlapping image-tiles instead of separate ones). In the present state, we do not consider the presented method ready to be used in clinical environment. However, we opine that the fuzzy approach and the set of antecedents used in the manuscript will find their place in this medical diagnosis field.
- There are important indices missing in the results, precision, NPV, MCC, Kappa and AUC.
We calculated these indices and included them in the manuscript. If all the 99 antecedents are used in the polyp detection, then these indices are not very promising, however, with advantageous selection applied to the antecedent-set the mentioned indices improve to usable levels.
- Proper feature selection must be used to show which features are informative. Statistical feature selection is preferred.
As the work reports on the present state of our experimentation with the fuzzy approach and with the set of antecedents, there are still other antecedents that are worth considering, we did not perform a full feature analysis and automatic selection of the features. We do appreciate this need though, and intend to use this widely studied area (e.g., based on singular value decomposition) in our future work.
- Due to the correlation between frames, and the importance of classifying new frame in a new subject, leave one out subject cross validation is required.
The database contains similar frames, but they do not follow each other in space and time too closely. In this way, the image tiles can be quite different. This remark – together with our previous good experience with the Rényi entropy-based structural entropies' – convinced us that using overlapping image tiles in the training phase could be a way forward, and could significantly improve the results, precisions and other characterizing indices.
- Comparison with the state of the art is missing.
We included a comparison of our results with those of the recent colonoscopy challenges in Subsection 4.4.
- Sensitivity analysis of free parameters is required.
It is a valid point and we intend to address it in a longer run. We intend to work out how to properly measure the uncertainty of the calculated antecedent parameters, however, as we have had no information about the uncertainty of the pixel intensities, therefore, we could not perform the sensitivity analysis.
Extensive overhaul of English has been carried out to improve the linguistic quality of our manuscript.
Reviewer 2 Report
Application of structural entropy and spatial filling factor in colonoscopy image classification
The authors described a fuzzy method and structural entropy (Rényi entropies) and spatial filling factor in colonoscopy imagery. The performed the automatic segmentation of the polyps and tested in available polyp images.
In my opinion, this manuscript has these good points:
- The subject is interesting and might absorb many readers in the field, I personally like the fuzzy logic to advance the project;
- Adequate analytical representations;
Also there are some suggestions that would increase the strength of the paper which is listed bellows;
- One of my major points about your article despite interesting topic concerns the highlighting novelty of this article, authors should highlight their contributions. I would recommend a workflow figure to show better your work.
- Authors mentioned: “In the following considerations first a fuzzy rule generation based on the statistical analysis and histograms of the antecedents is shown.” Please describe how you reached to fuzzy membership functions from statistics.
- Please provide your fuzzy rules in a table to be highlighted.
- Please cite: “Zadeh t-norm and s-norms”.
- I would strongly recommend considering to simplify the equations and make the entire article more interesting for readers by adding more descriptions for equations and more figures.
Some editorial points:
- Please enhance the quality of your Figures and Tables; The current figures are really nice but can be little bit more arranged.
- Please send your manuscript to an official proofreading to increase the readability of your manuscript;
In my opinion, this work is a very nice to be published after some improvements.
Thank you
Author Response
Response to the comments of Reviewer 2
Firstly, we would like to express our sincere thanks to Referee 2 for his/her insightful comments. We do hope that we managed to improve the quality of our manuscript via addressing his/her concerns.
The following points and concerns were addressed in the new version of our manuscript.
- One of my major points about your article ‒ despite its interesting topic ‒ concerns the highlighting novelty of this article, the authors should highlight their contributions. I would recommend a workflow figure to show better your work.
We have now included in the manuscript a workflow diagram (Figure 6.) of the developmental and processing steps required by and related to the proposed image-based colorectal polyp detection method. The method aims to assist medical experts in their tedious work of locating problematic areas in colonoscopy images and image sequences. Furthermore, with reference to the mentioned diagram, we have highlighted where we think our scientific contributions lie.
- Authors mentioned: “In the following considerations first a fuzzy rule generation based on the statistical analysis and histograms of the antecedents is shown.” Please describe how you reached to fuzzy membership functions from statistics.
We generated a number of histograms for image features ‒ see the detailed list of these features in Table 5. of our manuscript ‒ based on the images within the training set, normalized these histograms along both axes, and searched for the highest point of the histogram to be the core of the fuzzy rule. The support of the rule was determined by locating the first point where the histogram reached the 10% of its maximum value (starting from both sides).
- Please provide your fuzzy rules in a table to be highlighted.
The Appendix has been extended and it now includes a Table A1, Table A2 and Table A3 presenting the infimum, and supremum of the support and the core point for each of the 99 HSV colour space antecedents for both the positive and the negative medical outcomes.
- Please cite: “Zadeh t-norm and s-norms”.
We are very sorry to have overlooked to mention the Zadeh’s t- and s-norms. These are now properly referenced in the Subsection 3.1.
- I would strongly recommend considering to simplify the equations and make the entire article more interesting for readers by adding more descriptions for equations and more figures.
We have omitted the equations for the Canny filter, and corrected the equations describing the KH fuzzy rule-interpolation. We have re-written the figure captions in a more informative way. Furthermore, included some new figures in our manuscript to make it more interesting and to provide a feel for the application area and the detection task.
Some editorial points:
- Please enhance the quality of your Figures and Tables; The current figures are really nice but can be little bit more arranged.
The figures are printed now using larger fonts. Many of the figures have been simplified for better readability and understandability. Also, an informative figure about the histograms has been included in the Appendix.
- Please send your manuscript to an official proofreading to increase the readability of your manuscript;
Extensive overhaul of English has been carried out to improve the linguistic quality of our manuscript.
Round 2
Reviewer 1 Report
Most of the comments were implenented by the authors.
However, the following issues must still be taken into accounts,
- The CI 95% of the entire indices must be provided.
- As some of the indices are low, such as Kappa and mcc, their interpretation must be added to the discussion as limitations.
Author Response
Firstly, we would like to express our sincere thanks to Referee 1 for his/her insightful and helpful comments. We do hope that we managed to improve the quality of our manuscript via addressing his/her concerns.
The following points and concerns were addressed in the new version of our manuscript.
-
The CI 95% of the entire indices must be provided.
We calculated the indices and included them in the manuscript.
The CI95% confidence intervals were calculated by randomly selecting approximately 80% of the test set elements. 100 independent calculations were carried out for testing the confidence.
-
As some of the indices are low, such as Kappa and mcc, their interpretation must be added to the discussion as limitations.
The resulting indices are not too high, so our research results are not applicable for real image segment classification, so it is clear, that a further search for antecedents is needed, as well as a systematic optimization of the antecedents to be included, probably some bacterial or other nature-inspired algorithms, this paper is only the first step on the road where we studied, whether it is worth to start this direction, and whether the Rényi entropy-based quantities are applicable for this purpose.
Reviewer 2 Report
I have no comments. Congratulations!
Author Response
We would like to express our thanks to Referee 2 for his/her comment.
Round 3
Reviewer 1 Report
When looking at the results with CI 95%:
"MCC=0.19 ::::::: (CI95% [:::::: 0.1850,:::::: 0.1864]:: ),::::::::: PPV=0.33::::::: (CI95%: [:::::: 0.3253,:::::: 0.3261]: ),:::::::::: NPV=0.83::::::: (CI95%349 [:::::: 0.8338,:::::: 0.8343]:: ), :::::: κ=0.17::::::: (CI95% [:::::: 0.1719,:::::: 0.1732]: ).:::: The:::::: CI95%:::::::::: confidence::::::::: intervals::::: were ::::::::: calculated::: by350
::::::::: randomly:::::::: selecting:::::::::::::: approximately :: 80:: %::: of::: the:::: test::: set::::::::: elements.:::: 100:::::::::::: independent::::::::::: calculations::::: were351
:::::: carried:::: out ::: for:::::: testing::: the::::::::::: confidence. "
The value of the index must be within CI 95% [low, high], which is not the case in this paper, for example:
PPV=0.33 [0.325, 0.326] ... This occurs in almost all cases.
Please correct such issues which are not acceptable in any research area.
The authors must use a proper statistical formula to calculate the CI 95%, not to have this problem.
Author Response
Firstly, we would like to express our appreciation to Referee 1 for
his/her valid comments. We hope that now we have managed to address
his/her concerns.
As we had given the results originally with two-decimal-digit numerical
precision, after the previous change we did not change this numerical
precision as we thought it was good enough.
However, the confidence intervals were visible only in less significant
decimal digits.
In the present manuscript-version, we have corrected this discrepency
and have given the results concerned with four-decimal-digit numerical
precision.
Also, we have included in the manuscript the formula for computing the
confidence intervals.